pandasPGS: a Python package for easy retrieval of Polygenic Score Catalog data

Zhang Zheyu
Zhou Jintong
http://orcid.org/0000-0003-4308-1878 Cao Tianze tianze-cao@hznu.edu.cn
http://orcid.org/0000-0001-9298-5342 Huang Yuexia yxhuang@hznu.edu.cn
Huang Chu
Xia Yu
School of Mathematics, Hangzhou Normal University , Hangzhou, Zhejiang , China
Karakülah Gökhan
Electronic publication date: 2025 Feb 12
Publication date: 2025
Volume: 13
Electronic Location ID: e18985
Received 2024 Oct 15; Accepted 2025 Jan 22
Copyright: © 2025 Zhang et al.
Copyright year: 2025
Copyright holder: Zhang et al.
License: This is an open access article distributed under the terms of the Creative Commons Attribution License, which permits unrestricted use, distribution, reproduction and adaptation in any medium and for any purpose provided that it is properly attributed. For attribution, the original author(s), title, publication source (PeerJ) and either DOI or URL of the article must be cited.
License URL: https://creativecommons.org/licenses/by/4.0/

Keywords: Database, GWAS, PGS, Python, Data frame

Funding: Zhejiang Provincial Natural Science Foundation LZ23A010002 National Training Program of Innovation and Entrepreneurship for Undergraduates of Hangzhou Normal University 202410346057 This work was supported by the key project of Zhejiang Provincial Natural Science Foundation under grant number LZ23A010002 and the National Training Program of Innovation and Entrepreneurship for Undergraduates of Hangzhou Normal University under grant number 202410346057. The funders had no role in study design, data collection and analysis, decision to publish, or preparation of the manuscript.

==============================
Background

The Polygenic Score (PGS) Catalog is a public database dedicated to storing polygenic risk scores. To date, the database has included 5,022 polygenic risk scores associated with 656 different traits. Although the PGS Catalog offers an official resource representational state transfer (REST) application programming interface (API), there is no ready-made data client tailored for any specific programming language. Researchers are thus required to invest time in becoming familiar with the structure of the REST API and to implement a corresponding client in their programming language of choice to integrate PGS data into their analytical workflows.

Methods

In this work we introduce pandasPGS, a Python package that provides programmatic access to PGS Catalog data. After being called by the researcher, pandasPGS will automatically select the appropriate uniform resource locator (URL) and request the data based on the name and parameters of the called function, and merge the obtained pagination data. In addition, pandasPGS also provides further data pre-processing functions. According to the structure of the obtained data, it can convert the data into several hierarchical pandas.DataFrame objects, which is convenient for further analysis by researchers.

Results

This tool allows researchers to easily analyze PGS Catalog data using Python. It alleviates the time cost for researchers to learn the REST APIs of PGS Catalog. The source codes can be found in https://github.com/tianzelab/pandaspgs, and the API documentations can be found in https://tianzelab.github.io/pandaspgs/.

Introduction

Over the past decade, based on whole-genome association studies, researchers have discovered many traits influenced by polygenic associations. Polygenic risk scores are weighted numbers derived from whole-genome association studies that can predict susceptibility to certain traits based on genetic variations. The PGS Catalog (Lambert et al., 2024a), as the only publicly accessible authoritative database established by an authoritative institution, will attract more and more researchers to use its data for research as the number of included data increases.

In order to facilitate researchers to perform automated analysis, PGS Catalog provides data in JavaScript Object Notation (JSON) format through the resource representational state transfer (REST) application programming interface (API) (Lambert et al., 2024a). These JSON data are divided into nine categories according to different structures. For each specific category of data, the REST API provides several URLs and corresponding filtering parameters. It is necessary to select qualified URL based on specific filtering parameters to obtain data correctly.

The returned JSON data has a python dictionary-like structure, where the value of the key “results” is an array of the requested data. This array contains data for 0 or more specified categories based on the filter parameters. The maximum size of this array is controlled by the Hypertext Transfer Protocol (HTTP) request parameter named “limit” with 50 as the default value. For situations where there are more than 50 data that match the filter parameters, we need to use the URL with key “next” in this dictionary-like structure to request the remaining data. The server will then return JSON data with a similar structure. When the remaining data that matches the filtering parameters exceeds 50, the array with key “results” returned at this time will only contain the 51st to 100th data. Using the URL with key “next”, we can continue to request the remaining data. If there are more than 150 pieces of data or more, we need to repeat this process. Developers need to manually merge the arrays with key “results” in the above JSON structures to obtain the data of all composite filtering parameters. If the programming language used by the researcher does not provide a package that can retrieve this kind of data, then he will have to spend a lot of time to familiarize himself with the usage of the PGS Catalog API.

With the rise of deep learning, Python has undoubtedly become the most popular programming language. The PGS team has open sourced a Python package called pygscatalog (Lambert et al., 2024b). However, its main use is to query, download, and integrate scoring files from the command line interface. To make matters worse, the APIs are scattered across submodules in multiple three-tier folders, some lack parameter explanations, and don’t have a complete API programming example guide. As a result, it can be difficult for first-time users to find the right way to use these APIs. All of this makes it inconvenient for users to write Python programs based on it. There is still a lack of a Python tool that provides out-of-the-box data integration functions for PGS Catalog. It is necessary and urgent to develop a Python tool that meets the above functional requirements. This tool can effectively integrate PGS Catalog data into the current Python-based analysis pipeline, and allows users to perform automated programming-based analysis work.

Materials and Methods

Retrieving data from server

pandasPGS allows users to obtain the PGS Catalog by calling the functions get_cohort(), get_ancestry_categories(), get_performances(), get_publications(), get_releases(), get_sample_sets(), get_scores(), get_traits(), get_child_traits() and get_trait_categories(). pandasPGS is based on the cachetools package (Tkem, 2014), so while it obtains data, it also caches the data to reduce access to the server. The data cache validity time in pandasPGS is set to 24 h. If the data is not accessed within 24 h, its cache will be invalid. When the requested data is not cached, pandasPGS will assemble the URL for the data request based on the specific get_*() function and the passed parameters, and then request the data from the PGS catalog server based on the PGS Catalog REST API. When the returned data is paginated, pandasPGS will retrieve all paginated data in sequence and splice them together. pandasPGS will convert the obtained data into predefined objects. pandasPGS have defined two conversion modes. When the user requests data based on the “Thin” mode, pandasPGS will store the data in the attribute of raw_data. When the user requests data based on the “Fat” mode, pandasPGS will convert the data of the attribute raw_data into multiple DataFrames with hierarchical relationships based on the class structure. For ease of demonstration, five DataFrames are used in the figure (Fig. 1). The actual number will depend on the structure of the data. pandasPGS allows users to store data as CSV and EXCEL files respectively by calling the functions write_csv() and write_excel().

Figure 1 The main workflow of pandasPGS.

Convenient set operations

In order to facilitate users to organize data, pandasPGS provides some convenient set operation functions: bind(), union(), intersect(), set_xor() and set_equal(). In order to reduce the amount of coding for users, pandasPGS also provides the following mathematical symbols to support set operations: +(bind), &(intersect), −(set_diff), ^(set_xor), |(union), ==(set_equal).

Helper functions for accessing web links

In order to facilitate user browsing, pandasPGS allows users to directly open the corresponding web page by calling functions open_sample_set_in_pgs_catalog(), open_publication_in_pgs_catalog(), open_score_in_pgs_catalog(), open_trait_in_pgs_catalog(), open_in_dbsnp() and open_in_pubmed() with the corresponding identifier as the input parameter.

Structure for class score and other predefined classes

The class Score is defined in pandasPGS, which has two modes. In “Thin” mode, class Score contains two attributes: mode and raw_data. In “Fat” mode, class Score contains nine attributes: raw_data, scores, samples_variants, samples_variants_cohorts, trait_efo, samples_training, samples_training_cohorts, ancestry_distribution and mode. If the mode is set to Thin, the Score class will store the data in the attribute raw_data when constructing the object. If the mode is set to Fat, the Score class will also copy the data of the attribute raw_data and reprocess it, thereby generating the remaining seven attributes, which are all of the pandas.DataFrame type (McKinney, 2010).

The attribute scores is the main DataFrame. The attributes samples_variants, samples_training, trait_efo and ancestry_distribution are attached to the attribute scores. These attributes are related to the column id in the attribute scores through their respective score_id columns. Each row of data in the attribute scores has a one-to-many relationship with the data in these attributes.

The attribute samples_variants_cohorts is attached to the attribute samples_variants. They are related to each other through the column id in samples_variants and the column sample_id in samples_variants_cohorts. Each row of data in the attribute samples_variants has a one-to-many relationship with the data in the attribute samples_variants_cohorts. Likewise, a similar relationship exists between the attribute samples_training and the attribute samples_training_cohorts, and between the attribute samples_variants and the attribute samples_variants_cohorts.

The design of the Score-like attributes is to use attributes of type DataFrame as tables in a relational database, and some of the columns can be used as primary keys and foreign keys in the relational database. For example, the column id of the attribute scores is used as the primary key, and the score_id in other attributes is used as the foreign key. This allows the entity-relationship diagram in a relational database to well reflect the column names in the attributes and the relationships between them (Fig. S1).

Each attribute of other predefined classes in pandasPGS is also constructed based on the same design concept as the Score class (Figs. S2–S9).

Results

Example 1. Investigating trends in diabetes-related polygenic risk scores

pandas.DataFrame is a commonly used data type in pandasPGS. It is also the cornerstone of the Python data analysis ecological chain. This makes pandasPGS very easy to cooperate with other Python tools to complete data analysis work. This part will demonstrate the use of pandasPGS in conjunction with plotnine (Has2k1, 2017) to investigate the polygenic risk score associated with the trait named “diabetes”. plotnine is a Python implementation of ggplot2 (Wickham, 2016). The steps are as follows (code and console output are shown in File S1):

Step 1. Import the functions and classes from pandasPGS.

Step 2. Use the get_traits() function from pandasPGS to retrieve the trait data related to “diabetes” from the PGS Catalog and assign them to the variable traits. The variable traits is an instance of the pandasPGS.Trait class, which has six attributes that are pandas.DataFrame objects. Among them, the traits attribute is the main attribute, and the other five attributes are all associated with it.

Step 3. The identifier (id column) of the trait related to “diabetes” can be queried through traits.

Step 4. Use the get_scores() function of pandasPGS to query the trait-related Score data in Step 3 in sequence. The addition operation provided by pandasPGS is used to bind multiple pandasPGS.Score type data to the variable diabetes_score of type pandasPGS.Score.

Step 5. plotnine plotting uses pandas.DataFrame as the data source, and uses addition operations to overlay layers. Based on the data obtained from pandasPGS and the API provided by plotnine, the frequency distribution chart of Score data classified by traits can be easily drawn. It can be seen from the figure that the polygenic risk score related to “type 2 diabetes mellitus” is the most studied trait categorized under “diabetes” (Fig. 2).

Figure 2 Statistical distribution of the number of polygenic scores for diabetes-related traits.

Example 2. Investigating polygenic risk scores for gestational diabetes

Gestational diabetes may cause the fetus to grow extremely quickly, increasing the possibility of premature delivery and dystocia. In addition, gestational diabetes may also induce other pregnancy complications, such as gestational hypertension. It may also cause severe trauma to the birth canal after delivery, such as postpartum hemorrhage. This part will demonstrate how to use pandasPGS to generate corresponding genotypes and scores. The steps are as follows (code and console output are shown in File S2):

Step 1. Import the functions and classes from pandasPGS.

Step 2. Use the function get_traits() to query data related to “gestational diabetes”. pandasPGS will return a pandasPGS.Trait object and assign it to the variable traits. Its attribute traits is its main DataFrame. The attribute traits is the main DataFrame, and its summary information can be obtained by printing. Use the square bracket operator to index the data of the corresponding cell. By indexing the column id and column description in row 1, we can query that the identifier of the corresponding trait is “EFO_0004593” and the corresponding description is “Carbohydrate intolerance first during diagnosed pregnancy. (NCIT: P378)”.

Step 3. Given the identifier of the corresponding trait, pandasPGS can query the corresponding pandasPGS.Score type data through the function get_scores(). The attribute scores is the main DataFrame, and its summary information can be obtained by printing. The value of column id is the identifier of its pandasPGS.Score, and column name is the name of the corresponding polygenic risk score. The value of the matches_publication column is True, which means that the polygenic risk score has been published in a article. The value of column trait_reported describes how the corresponding trait is named in the article (Wu et al., 2022). The value of column variants_number is 4, indicating that the polygenic risk score is composed of four variants.

Step 4. Metadata information for published PGS articles is stored in columns with names begin with “publication”. For example, “publication.id” stores the identifier of this article in PGS Catalog, and “publication.PMID” stores the identifier of this article in PubMed.

Step 5. For understanding the applicability of this polygenic score, it is important to understand the samples used to define the variant associations (effect sizes) used in PGS. Such information is stored in the attribute samples_variants. In this table, we can see a sample of 671 individuals used in the Genome-Wide Association Studies (GWAS) stage (Buniello et al., 2019). The ancestry category associated with these individuals is East Asian. Based on this information, one could hypothesize that the applicability of this risk score should probably be limited to people of East Asian ancestry.

Step 6. Use read_scoring_file() to download a PGS file from the PGS Catalog and convert it into a pandas.DataFrame. It can be seen from the table that the corresponding distributions of the four variants are rs10830963, rs1436953, rs7172432 and rs16955379. The corresponding chromosome and position data in the chromosome are recorded in columns hm_chr and hm_pos. The effect allele and other allele are recorded in columns effect_allele and other_allele. The weight corresponding to the gene in the column effect_allele is recorded in the column effect_weight.

Step 7. Taking rs10830963 as an example, we calculate its corresponding genotype and corresponding weighted score. pandasPGS provides the function genotype_weighted_score() for this purpose. The genotypes of rs10830963 are G/G, G/C and C/C respectively. The weighted score of G/G is 1.327×2=2.654. The weighted score of G/C is 1.327×1=1.327. The weighted score of C/C is 1.327×0=0.

Step 8. Based on the calculation process of Step 7, one can use loops to calculate the genotypes and weighted scores of the four variants. Then one can use the function reduce() and the method merge() of the object pandas.DataFrame to calculate a DataFame object combination_df, which lists all combinations of genotypes composed of four variants, as well as the corresponding weighted scores.

Step 9. The data of column genotypes is obtained by splicing together the genotype combinations of four variants. The column scores is the sum of the corresponding four weighted scores. Based on the results of descending sorting on the column scores, we can see that the range of scores is 0 to 10.244. Potential patients can assess their risk based on their genotype combination.

Discussion

pandasPGS vs. quincunx

Among existing tools, quincunx (Magno, Duarte & Maia, 2022) is the only one with similar functionality to pandasPGS. It is written in R language and can be integrated into the current R language data analysis set. pandasPGS is more functionally complete than quincunx (as shown in Table 1).

Table 1 pandasPGS vs. quincunx.

	pandasPGS	quincunx	
Programming language	Python	R	
Mode	Fat or Thin	Fat	
Type of attribute	pandas.DataFrame	tidyverse.tibble	
Cache management	Classified	Not classified	
Set operations	set_xor, bind, union, intersect, set_diff, set_equal	bind, union, intersect, set_diff, set_equal	
Set operations based on mathematical symbol	+(bind), & (intersect), −(set_diff), ^(set_xor), |(union), ==(set_equal).	Unsupported	
The mapping between the columns of the table and the key of JSON in the PGS Catalog REST API	Strong	Weak	

First, pandasPGS provides two working modes: Fat and Thin. In Fat mode, pandasPGS will convert the obtained JSON data into pandas.DataFrame objects. In Thin mode, pandasPGS acts like a pure client, merging paginated JSON data and converting it into a Python list.

Second, pandasPGS has more sophisticated cache management. Cache data is divided into nine categories according to its JSON data type. When necessary, only one or several categories of caches can be cleared, while in quincunx all caches are cleared directly.

Third, although both quincunx and pandasPGS support set operations on the acquired data, quincunx does not support the set_xor() method. In addition, pandasPGS also supports the use of mathematical symbols instead of function calls to simplify set operations.

Finally, pandasPGS strictly names the column names of the converted DataFrame according to the key names of the JSON data provided by the PGS Catalog API, but quincunx does not do this. For example, the official documentation has data with the key variants_genomebuild in a structure with JSON type Score. In pandasPGS this data is stored in the variants_genomebuild column of a DataFrame, but in quincunx the column name for this data is assembly. Users may be confused by the inconsistent naming between official documentation and client tools.

Experiments to obtain full data for data conversion

In order to test the ability to integrate PGS Catalog data, we used pandasPGS and quincunx respectively to obtain all types of data in PGS Catalog and convert them into corresponding data frames (code and console output are shown in File S3). The test results show (as shown in Table 2) that pandasPGS can successfully obtain all data and convert it into the corresponding data frame. quincunx failed to complete due to a server exception triggered when retrieving data of type Cohort. quincunx also failed because there was no compatible data type when obtaining data of type PerformanceMetric and SampleSet, as well as Score data. Compared with quincunx, pandasPGS optimizes the parameters of HTTP communication with the PGS Catalog server, which reduces the server overhead and makes it less likely to trigger exceptions. pandasPGS is also developed based on the latest release of PGS Catalog and has been thoroughly tested to ensure compatibility with data formats.

Table 2 Comparison of compatibility capabilities of all data conversions.

Type	pandasPGS	quincunx	
AncestryCategory	Success	Success	
Cohort	Success	Fail	
PerformanceMetric	Success	Fail	
Publication	Success	Success	
Release	Success	Success	
SampleSet	Success	Fail	
Score	Success	Fail	
TraitCategory	Success	Success	
Trait	Success	Success	

Cache management

The PGS Catalog server has two restrictions on the REST API. First, a single IP address can request data up to 100 times per second. Second, if there are more than 50 data requested, the data will be paginated at 50 per page. The user must follow the URL provided by the API with pagination parameters to get the remaining pages. pandasPGS provides data caching to mitigate the impact of these two restrictions. Caching can prevent the program from requesting the same data in a short period of time and triggering the first restriction, thus avoiding IP blocking. The second restriction has an impact on users with an unstable network connection to PGS Catalog’s servers, as the program may fail due to intermittent network outages. The cache retains the paging data that has already been retrieved in the event of a network outage, allowing the program to quickly recover from failures.pandasPGS prepares an adjustable cache pool for each get_*() function. Each cache pool can store data for up to 2,048 recent requests and is valid for 24 h. If there is a 2,049 item of data that needs to be saved and none of the data in the cache pool has expired, the cache pool will delete the data that has not been used for the longest time based on the least recently used (LRU) principle. Each get_* function has a parameter called “cached”. When the parameter “cached” is set to False, pandasPGS will forcibly request the latest data from the PGS Catalog server and store the results in the cache pool. pandasPGS provides function reinit_cache() for reinitializing and configuring the maximum capacity of the cache pool and data validity time, and function clear_cache() for emptying the cache pool data.

Conclusions

pandasPGS is developed according to the latest release of PGS Catalog. It can obtain PGS Catalog data that meets the filtering conditions based on the input filtering parameters and convert it into the corresponding data frame with hierarchical relationships. Compared with other existing tools, it is the only one that can obtain the full PGS Catalog data and convert them into data frames.

Supplemental Information

Supplemental Information 1 Code used in Example 1.

The file contains the code and the main program output. The program output starts with "#".

Supplemental Information 2 Code used in Example 2.

The file contains the code and the main program output. The program output starts with "#".

Supplemental Information 3 Code of experiments in data retrieval and transformation.

The file contains the code and the main program output. The program output starts with "#".

Supplemental Information 4 Structure for class Score.

Supplemental Information 5 Structure for class Cohort.

Supplemental Information 6 Structure for class PerformanceMetric.

Supplemental Information 7 Structure for class Publication.

Supplemental Information 8 Structure for class Release.

Supplemental Information 9 Structure for class SampleSet.

Supplemental Information 10 Structure for class Trait.

Supplemental Information 11 Structure for class TraitCategory.

Supplemental Information 12 Structure for class AncestryCategory.

Supplemental Information 13 Data from variables in Example 1.

All files are in CSV format. The folder and file names match the variable names in the code.

Supplemental Information 14 Data from variables in Example 2.

All files are in CSV format. The folder and file names match the variable names in the code.

This work was inspired by quincunx. The authors would like to thank the developers of quincunx.

Additional Information and Declarations

Competing Interests

The authors declare that they have no competing interests.

Author Contributions

Zheyu Zhang performed the experiments, analyzed the data, prepared figures and/or tables, authored or reviewed drafts of the article, and approved the final draft.

Jintong Zhou performed the experiments, analyzed the data, prepared figures and/or tables, authored or reviewed drafts of the article, and approved the final draft.

Tianze Cao conceived and designed the experiments, performed the experiments, analyzed the data, prepared figures and/or tables, authored or reviewed drafts of the article, and approved the final draft.

Yuexia Huang conceived and designed the experiments, authored or reviewed drafts of the article, and approved the final draft.

Chu Huang conceived and designed the experiments, authored or reviewed drafts of the article, and approved the final draft.

Yu Xia conceived and designed the experiments, authored or reviewed drafts of the article, and approved the final draft.

Data Availability

The following information was supplied regarding data availability:

pandasPGS data is available at Zenodo:

CaoTianze, & zzystc66. (2025). TianzeLab/pandaspgs: The second released version. (v0.2.0). Zenodo. https://doi.org/10.5281/zenodo.14619537.

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
