# Peer review of "pandasPGS: a Python package for easy retrieval of Polygenic Score Catalog data"

_PeerJ, doi:10.7717/peerj.18985_

## Round 0.1 · original submission · Minor Revisions

Both reviewers agree that your manuscript meets the required standards for publication; however, they have suggested some improvements that need to be addressed before acceptance. I kindly invite you to revise your manuscript in line with the reviewers’ comments and resubmit for further evaluation.

Reviewer 1 ·

Basic reporting

The manuscript was written in a professional structure using clear and professional English. Sufficient background was provided. Source code, examples, and document were included.

Experimental design

I have main concerns related to the cache:

1. The cache validity is strictly set to 24 hours. If a user accesses the data slightly beyond the 24-hour window, the cache becomes invalid, forcing a new data request. This could lead to unnecessary delays and redundant server requests, especially for infrequently updated data. It would lead into Increased latency and potential overloading of the PGS server for data that hasn't changed.

2. While caching reduces server requests, users relying on the cache might work with data that is up to 24 hours old. This is problematic if the PGS catalog updates frequently and users need the most current information. So, analyses or applications using outdated data might lead to incorrect conclusions or decisions.

3. Caching all accessed data for 24 hours could lead to significant memory usage, especially for large datasets or if multiple users access varied data through pandasPGS. It may lead to reduced performance or crashes in resource-constrained environments.

4. If the cache is not accessed frequently within the 24-hour window, it will expire and need to be refreshed again, potentially wasting resources to store unused data. It might lead to wasted computational resources for caching low-priority or seldom-used data.

Please take these potential issues into account. I recommend to add contents related to cache into discussion part.

Validity of the findings

The python package is powerful. Here, I just provide some potential solutions to resolve the cache potential issues so that the caching system can become more robust, flexible, and efficient. Also, these can be added to discussion section.

1. Customizable Cache Lifetime: Allow users to adjust the cache validity time based on their needs or the expected update frequency of the PGS catalog.

2. Data Change Notifications: Integrate mechanisms (e.g., through the PGS API) to detect changes in the catalog and invalidate affected cached data.

3. Selective Caching: Provide options to cache only specific data subsets or prioritize frequently used datasets.

4. Manual Cache Refresh: Allow users to refresh the cache on demand for specific queries without waiting for expiration.

Additional comments

NA

Reviewer 2 ·

Basic reporting

The authors have written a nice application note in the field of polygenic scoring. The tool will be very helpful for people that would like to query the PGC catalog programmatically.

The level of English language use is good. Background and context is sufficient.

The article is a good read and is structured nicely.

As this is an application note, it is hard to establish if the results are appropriate in the context of hypotheses. So, it is not possible to tale this criterium into account.

Experimental design

Again, this is an application note.
As a software application, it seems well-structured. The code is open sourced and is documented. Functionalities are well thought-through.
The authors have compared their python module to its inspiration: an R library. This package performs similar tricks, but has a lower level of functionality.

Validity of the findings

I have checked the provided example codes and they worked properly on my machine. Installation went as described in the documentation.

Additional comments

I do not know if the application note publication type is within the scope if the intended PeerJ journal. Other than that, I'm fine with publishing after the minor points raised below:

* Lines 20-21: Maybe update the numbers. When I accessed the PGS catalog, they were already higher than in the manuscript.
* Line 36: change "documentations" to "API documentations"
* Lines 42 and 43: Change "diseases" to e.g. "traits". PGSs are not only used for diseases, but for many other phenotypes as well.
* Lines 53-54: I don't understand what you mean at the end of the sentence "...the dictionary item corresponding to result as the key." Could you please rewrite the sentence such that it becomes clearer?
* Line 71: "very complicated" is not subjective. Please rephrase.
* Lines 170-171: Please change "... is a 171 popular research direction (Fig. 2)." to "... is the most studied trait categorized under "diabetes".".

---

## Round 0.2 · accepted · Accept

Based on my assessment as an Academic Editor, the authors have adequately addressed the reviewers’ concerns. I find the manuscript to be satisfactory in its current form and recommend it for acceptance without further revisions.